# Validation of the Arabic Version of the Transition Planning Inventory (TPI-AR)

**DOI:** 10.3390/ijerph20021135

**Published:** 2023-01-09

**Authors:** Ghaleb H. Alnahdi, Arwa Alwadei

**Affiliations:** Department of Special Education, College of Education, Prince Sattam bin Abdulaziz University, Al-Kharj 11942, Saudi Arabia

**Keywords:** transition planning inventory, arabic TPI, psychometric properties, students with disabilities, transition planning, confirmatory factor analysis

## Abstract

The Transition Planning Inventory (TPI) is an important tool for planning the transition to life after school for students with disabilities. While interest in transition services has increased in the last decade in the Arab region, no transition assessment tools validated for the Arab population are currently available. This study is the first to validate an Arabic version (TPI-AR) for all three rating forms (student, home, and school) and examine its psychometric properties. The sample comprised 203 students with disabilities, a member of their family, and one of their teachers. The 11 subscales of TPI-AR for all three forms were found reliable and valid to be used with students with disabilities in Saudi Arabia, particularly in middle and high schools.

## 1. Introduction

In recent decades, medical and educational services for youth with disabilities in Saudi Arabia have witnessed significant improvement in terms of the provision of care and services to people with disabilities [1,2,3]. This improvement has targeted individuals with various disabilities and of different ages. Individuals in transitional stages have also been a part of this focus in terms of supportive regulations. In this period, an individual with a disability moves from a school stage to a post-school stage that has different requirements or moves to a stage of complete independence from a stage in which they depend on family or service providers. One example of this improvement can be seen in the attention given in the Special Education Regulatory Manual to providing a range of services to students in the transitional ages and preparing transition plans based on the capabilities and needs of each student to help them achieve coexistence and stability in after-school life [4].

Transition services have received great attention because they help people with disabilities to become aware of their abilities and needs and raise their efficiency and skills to improve outcomes [5]. Transition planning is one of the important ways to prepare students with disabilities for integration into society. The Individuals with Disabilities Education Act [6] also emphasizes the link between transition planning and positive transition outcomes for people with disabilities in after-school life. To be effective, transition planning should be based on appropriate assessments so that transition goals can be set according to students’ needs, strengths, and preferences. Thus, transition assessment serves as the basis for planning and providing transition services [7]. Despite the importance of formal and informal assessments of planning success [8,9], there is a significant shortcoming in their implementation in Saudi Arabia [10,11,12,13]. Over the last decade, there has been a small increase in the number of researchers interested in the field of transition services in the Arab region [14,15,16,17,18,19]. However, there is still a lack of studies on tools that may help improve transition practices in the Arab world, and Saudi Arabia in particular. Therefore, a lack of transition assessment tools and standards that have been validated for the Arab population is a standing issue.

This study attempts to bridge this gap by validating an Arabic version of the Transition Planning Inventory (TPI), a screening tool that has a theoretical basis and suitable psychometric properties and one of the popular assessment tools [9,20]. The TPI is “designed for all students receiving special education services who have reached the age that mandates transition planning” (p. 15), from 14 years old and older [21]. The TPI has been shown to have adequate psychometric properties to be a comprehensive assessment tool for planning transition services [22,23]. This is due to the need for further improvement in the quality of transition planning [10]. The scale is designed to provide a comprehensive list of basic information regarding students’ needs, strengths, and preferences to implement effective and well-informed transition planning [24,25]. It can assist students and their families in making decisions during the planning process for transition goals [26]. It can also enable individual educational program (IEP) teams to organize meetings based on TPI assessment results that are consistent with recommended practices in IEP planning for transition services and the achievement of transition goals [27,28]. The validity of the content, domains, and elements of the TPI is consistent with what the literature has emphasized regarding transition [29]. The TPI can thus be used by schools to formulate an effective transition program [23].

Several studies have confirmed the reliability of TPI items for the three respondents (students, parents, and school staff), as well as the reliability of retesting [21]. In addition, a Spanish version of the TPI (home form) was validated [30]. The TPI has also been used in studies to verify agreement and differences in ratings of strengths and transition needs among students, their parents, and school staff [31,32,33,34]. These studies have highlighted that views vary in some areas, and also emphasized the positive impact resulting from the presence of more than one evaluator in terms of collecting information from several points of view. Families offer the perspective of their children and their behavior in society, while teachers view them within the school walls. Studies have also used the TPI as a tool to assess transition domains such as independence skills [20,35] communication, interpersonal relationships, and self-determination skills [36].

The TPI has evolved in the previous decades from the first version [37] to the third version [21], passing through the second version [38]. TPI-1 [37] contains 9 domains in comparison with 11 domains (subscales) in the second and third versions. TPI-Second Edition (TPI-2) [38] measures three main areas of transition: work, learning, and living. These areas are further divided into 11 transition planning areas comprising 57 elements. It considers students’ interests, preferences, strengths, and needs. Information is collected from three assessors: the student, the parent, and the school employee. The instrument is used for students aged 14–21 years. 

The reliability of TPI-2 has been previously confirmed, with the internal consistency reliability showing a good reliability of >0.80 [39]. The alpha coefficients for the 11 planning domains were within the acceptable to good range. As each of the categories in the 11 plot areas had only 4 to 8 statements, it measured a limited amount of variance [21]. The reliability of the time samples was calculated using the test–retest method by sampling the test reapplied over an interval of 7 to 10 days with a sample of 55 students. The three main domains had correlations above 0.70, except for one coefficient that scored 0.65. Of the 33 coefficients for the 11 planning domains, 82% were above 0.70 and 18% were below 0.70. The lowest test–retest reliability coefficient was found among students’ ratings of the working broad area, and specifically for the career choice and planning subcategory. This is justified by the fact that students are in a stage of growth and may change their attitudes toward making decisions [40]. The validity of the criterion was tested by comparing the TPI-2 with the Enderle–Severson Transition Rating Scale. The results demonstrated a moderate to a high level of correlation between the two tools in the items of work, education, and living [21]. The validity of the content of TPI-2 was confirmed by field tests and understanding of the received comments, and the tool was found compatible with the IEP imposed by IDEA and addresses all the basics of transition planning. 

The new version (third) of TPI-3 released in 2021 is mainly the previous version, with some minor additions as supplementary tools in the TPI-3 kit, based on the evaluation results of the TPI-2 [21]. TPI-3 retained the same 11 domains and the same 57 items [21]. Therefore, the psychometric properties of TPI-2 were used and cited by developers as evidence of the good psychometric properties of TPI-3. TPI-3 was the version used for the current study.

In this context, the TPI scale has important characteristics that are useful for constructing transitional plans. As there are no alternatives to the TPI in the Arabic language, this study contributes by aiming to provide a validated Arabic version of the scale (TPI-AR) that can help create more comprehensive and well-directed plans to prepare youth with disabilities for work and adulthood. In sum, this study aims to answer the following question: Is the Arabic version of TPI-AR a valid and reliable instrument to be used in Saudi Arabia?

## 2. Method

### 2.1. Participants

Before collecting data, permission was obtained from the publisher of the TPI for translating and validating an Arabic version of the scale. Approval from the Institutional Review Board at the university was also obtained. The study sample comprised 203 students from intermediate and secondary schools. This sample is considered sufficient for confirmatory factor analysis, especially with no missing responses, as it is larger than 150 [41]. Around 66% of the participants had intellectual disabilities, 15% had hearing impairments, 11% had learning disabilities, and 8% had visual impairments. Of the participants who filled out the TPI home form, around 49% were mothers, 22% were fathers, 20% were sisters, 7% were brothers, and the rest were other relatives. About 80% of the sample who filled out the school form had a bachelor’s degree, 16% had a graduate degree, and 3% had a diploma in special education (see Table 1). 

### 2.2. Data Collection

As this was the first time that the TPI was used in the Arabic language, the scale was translated from the English version by following the standards proposed by Beaton et al. [42]. The items were translated into Arabic by three bilingual educators who were fluent in both Arabic and English. Then, the three translations were reviewed by two others (the authors) who combined the three versions into a unified version in the Arabic language. Next, the standard Arabic version was translated back into English by two different bilingual translators. After this, the translated English versions were compared with the original, and it was ensured that the original meaning of the items had been preserved. Then, the combined Arabic version was used in a pilot sample (*n* = 30) to ensure the clarity of the items and rectify any unclear items. After the application of the pilot sample, approval from the Ministry of Education office was taken to implement the scale. Next, after obtaining approval for the application of the TPI, direct contact was made with integration schools and daycare centers in the regions of Riyadh and Makkah to recruit participants for the study. Additionally, some teachers were willing to voluntarily participate in the application and data collection for this study. Then, we called for assistants (volunteers) to participate in the implementation of the scale among teachers from different regions. Researcher assistants (teachers) from Taif, Riyadh, and Al-Kharj responded and were willing to participate. Therefore, the sample of this study was dependent on whether the schools’ administrators and teachers were willing and had time to support the data collection process. Next, assistants were contacted individually. We explained the scale to them, and the authors followed up the process of applying the scale to the sample. The assistants’ part mostly was dedicated to distributing and collecting the home and school forms. The student form was applied personally by the second author. This was conducted in three different ways: face to face with students, communicating with them by phone, or through some online platforms such as Zoom. 

### 2.3. Data Analysis

The analyses in this study were conducted in three steps. First, data were checked to screen for outliers, missing responses, and descriptive statistics. Second, Cronbach’s alpha was computed to examine reliability. Third, multiple confirmatory factor analysis was conducted to examine the structural validity. Two main software packages were used to analyze the study data: IBM SPSS Statistics 21 and SPSS Amos 20 with a maximum likelihood estimation. 

## 3. Results

The first step in screening data was by checking for missing responses. There were no missing responses in the main items, while two teachers missed reporting the demographic information. Since the demographic data for the two teachers were not related to the confirmatory factor analysis, no action was needed. The next descriptive statistics were screened to check that all responses were recorded correctly within the acceptable range (from 0 to 5 for each item), as it goes from 0 to 5 based on the agreement level with each item. The results, presented in Table 2, show that the alphas ranged from 0.89 to 0.98 for the school form, 0.88 to 0.97 for the home form, and 0.83 to 0.97 for the student form. These values are a good indicator of the reliability of all subscales [43]. In sum, the three forms and all subscales of the TPI show very good indications of internal consistency. 

### Structural Validity

The original version of TPI-3 contains 11 dimensions in three main areas. After finishing the back translation according to the proposed guidelines [42], we chose to use confirmatory factor analysis (CFA), as it was stated that “to determine the structure of the instrument, a factor analysis is the preferred statistic when using CTT (Classical test theory)” [44] (p. 32). Given that the TPI “is supported from both theoretical and practical precatives” [21] (p. 3), using CFA is important to examine whether the psychometric properties found in the original version would be comparable to the Arabic version. In other words, the 57 items represent 11 dimensions on which the scale was validated in the original version [21]. It was expected that the first step for those who want to test another version of this scale is to verify whether these 57 items represent these 11 dimensions (loads significantly) as it is hypothesized. From here comes the role of confirmatory factor analysis to test this hypothesis. Three steps were conducted to examine the structural validity of the scale. We examined three models of structures via confirmatory factor analysis (CFA). For Model 1, it was assumed that the 11 subscales are independent scales, as each subscale measures a distinct variable [37]. Other researchers have also implied the same by using means on all subscales rather than an overall mean of all items [31,32,35,45,46]. In addition, there are researchers who have not used all subscales to measure the mean for these subscales [20,28,30]. Therefore, here, we are not examining any correlation among the subscales in this model.

In Model 2, three upper latent variables were assumed as the first two subscales representing working skills; the next three represented living skills, and the last six subscales represented living skills. Examining the fit of Model 2 would help in understanding if there is support to have an overall mean on these three second-order latent variables (working, learning, and living). For example, researchers and practitioners would be supported to calculate means of working, learning, and living. In the last model, Model 3, it was assumed that there is a third-order latent variable that would examine if there is support to have an overall mean of all items (57 items) in the scale. If Model 3 fits the observed data, that means that calculating a mean score on the 57 items would be supported. All three hypothetical models are shown in Figure 1.

Table 3 shows the fit indices for Model 1, where all 11 subscales in the TPI show indicators for good fit as distinct scales that each measure a distinct latent variable. For example, a value of less than 0.08 for the root mean square error of approximation (RMSEA) [47], a value higher than 0.9 for the comparative fit index (GFI) [48,49], higher than 0.9 for the goodness-of-fit index, and a value higher than 0.9 for the Tucker–Lewis coefficient (TLI) are all indicators for good fit. The fit indices in Table 3 show the criteria for fitting were met for Model 1, where all subscales were independent scales, and each subscale measures a distinct variable. The loadings of items on all 11 subscales were loaded significantly on each subscale (>0.4) [50], which can be considered as an indicator of the unidimensional scale that measures one underlying variable. For example, item loadings on Career Choice and Planning (latent variable) ranged from 0.7 to 0.89 for the students’ form, from 0.76 to 0.89 for the parents’ form, and from 0.76 to 0.95 for the teachers’ form. Another example, item loadings on Self-Determination (latent variable), ranged from 0.69 to 0.88 for the students’ form, from 0.74 to 0.93 for the parents’ form, and from 0.74 to 0.96 for the teachers’ form (see Appendix A for more details for all 11 subscales).

Next, we examined the fit indices for Model 2 with three upper latent variables (working, learning, and living). The results show that the model was not supported, as the fit indices did not meet the minimum criteria to indicate the observed data fit the hypothesized model [47]. For example, the goodness-of-fit index (GFI) = 0.440, adjusted goodness-of-fit index (AGFI) = 0.397, incremental measure of goodness of fit (NFI) = 0.592, and *X*^2^/df = 3.4. The third model did not show better-fit indices than the second model. For example, the GFI = 0.415, adjusted goodness-of-fit index (AGFI) = 0.372, SRMR = 0.0440, incremental measure of goodness of fit (NFI) = 0.550, and *X*^2^/df = 3.7. These fit indices values were not supportive of the third model. This means that calculating means on the overall scale (57 items) would not be recommended for the researchers and practitioners using TIP-AR. In addition, calculating the means of the three upper latent variables (working, learning, and living) is not statistically supported, as Model 2 did not fit the observed data. In sum, the first model with 11 separated subscales had the best-fit indices, which supported having a mean score on each of the 11 subscales as a unidimensional measure. In addition, this supports the possibility of applying subscales independently and according to the needs of any practitioner or researcher, since there is a base that each subscale represents an independent latent variable. 

The content validity (CV) was examined by 12 experts in related fields to provide services for people with disabilities [51]. The content validity was examined using four criteria, namely, relevance, clarity, simplicity, and ambiguity [52]. We used the content validity index (CVI) to quantify the level of agreement between the experts [51,52,53]. This was achieved by calculating the proportion of the experts who rate the item with 3 or 4 (for example, very relevant or quite relevant in the relevance criteria) [51]. The CVI ranged from 0.91 to 1 for all items for the four criteria. This can be considered as a good indicator of content validity. In addition, the results in Table 4 demonstrate the composite reliability (CR) values and average variance extracted (AVE) for all the subscales. Results in this table show good indicators for composite reliability with values equal to or higher than 0.7 [54]. In addition, indicators for convergent validity were obtained with values higher than 0.5 for AVE on all the subscales [54,55].

## 4. Discussion

This study was the first to devise and confirm the validity of an Arabic version of the TPI. Importantly, it confirmed the psychometric properties of the three rating forms of the TPI (student, home, and school). A comprehensive, reliable, and valid transition assessment tool is of paramount importance for developing quality transition plans for students with disabilities. The study examined the psychometric properties of an Arabic version of the TPI that was translated and analyzed using a sample from Saudi Arabia. The results show good indicators of the TPI and indicate that it is suitable for use in youth with disabilities in Saudi Arabia. TPI-AR is expected to help improve teacher practices in implementing good transition planning based on a comprehensive assessment of the needs and strengths of students with disabilities, consistent with the requirements of IDEA [20,23]. It is also expected to contribute to setting more realistic and appropriate transition goals to achieve effective transitional outcomes [28].

Having an Arabic version of the TPI would help schools to bring together multiple points of view (school, home, and student), thus providing a comprehensive perception about the students’ performance levels. It also fulfills the desires of families to participate in and express their aspirations and expectations about the students’ future and provides the student an opportunity to express their desires and needs, which raises their competence in self-awareness [34,46]. It can be used for all categories, including students with mild and moderate intellectual disabilities, with the exception of students with autism and severe disabilities. Such students have another form that emphasizes the importance of providing guidance and instructions to ensure their understanding of the scale [20,35]. The literature frequently emphasizes the importance of the IEP team transition assessment to take advantage of students’ strengths and preferences and consider their individual needs to prepare for areas of transition beyond school [23,56]. From here, and due to the availability of an Arabic version of the TPI, the assessment in Saudi schools should include the necessary specifications, such as determining the student’s ability for job engagement, community participation, interpersonal skills, and the level of self-determination, in addition to the means of entertainment and recreation. That is, it must comprehensively assess all domains related to the transitional areas of working, learning, and living [9]. Prior to this study, there was no comprehensive Arabic assessment tool for measuring the main areas of transition. The assessment tool developed in this study is comprehensive on all these counts and can help schools improve transition outcomes for students with disabilities.

### 4.1. Implications for Practice

Our main objective in this study was to address the shortcomings in implementing transition assessments for students with disabilities in Saudi Arabian schools. We addressed this by validating an Arabic TPI scale, which was a translated version of the TPI, one of the popular transition assessment tools. The main contribution of the current study is that it provides the first Arabic transitional assessment tool that schools can utilize in the transition planning process. This development is promising in the process of encouraging and training IEP teams to use it as an initial step to identify indicators about students’ current performance and their families’ aspirations for their children’s future. Specifically, IEP teams can use it in the initial stages of transition planning to identify indicators about students’ current performance and their families’ aspirations regarding their future. This Arabic version of TIP can be used for all special education programs for students with disabilities (except students with autism and severe disabilities) across Saudi Arabia. The availability of this Arabic version in schools will also help to direct the staff and teachers in the special education programs or in the centers that provide services for people with disability to emphasize and focus on preparing students for the post-school stage, especially with having a valid reference to measure the level of progress in this aspect.

### 4.2. Implications for Research

In addition, the study contributes by providing researchers with a valid and reliable tool that can be used to conduct further research. In future studies, Multitrait-Multimethod (MTMM) analysis will prove informative for investigating the convergent and discriminant validity of the TPI from the three resources and examining whether collecting data from the three resources brings additional understanding in comparison with two resources. One of the benefits of MTMM analysis is that it permits for assessing underlying trait factors from different sources (desirable) and whether there is an underlying (undesirable) method factor [57]. Bearing in mind that one of the requirements for using MTMM analysis is at least two methods and two traits [58,59,60], this applies perfectly to TPI data, where there is a possibility to have 11 traits (subscales) and three methods (resources; student, family member, and teacher).

### 4.3. Limitations

There are a couple of limitations that should be considered while interpreting the study findings. The first limitation of reading the results of this study is related to the fact that the sample is from Saudi Arabia only. The terms used in the instruments are easy and direct, and they are expected to be appropriate for other samples from other Arab countries. Issues related to the sample size and recruitment procedures might impact the generalizability of the findings. However, the sample size for the first and suggested model has no issue, as the subscales were analyzed independently as distinct scales, and the number of items does not exceed 20 items for any of the subscales. In addition, more studies from different parts of Saudi Arabia and from other Arab countries will help to confirm the study findings.

## 5. Conclusions

In conclusion, this was the first study to devise and validate the psychometric properties of TPI-AR. The results showed that it can be a useful tool for middle and high schools in Saudi Arabia. Its availability will help researchers and practitioners who are interested in a tool whose psychometric properties have been tested on different samples from different populations.

## Figures and Tables

**Figure 1 ijerph-20-01135-f001:**
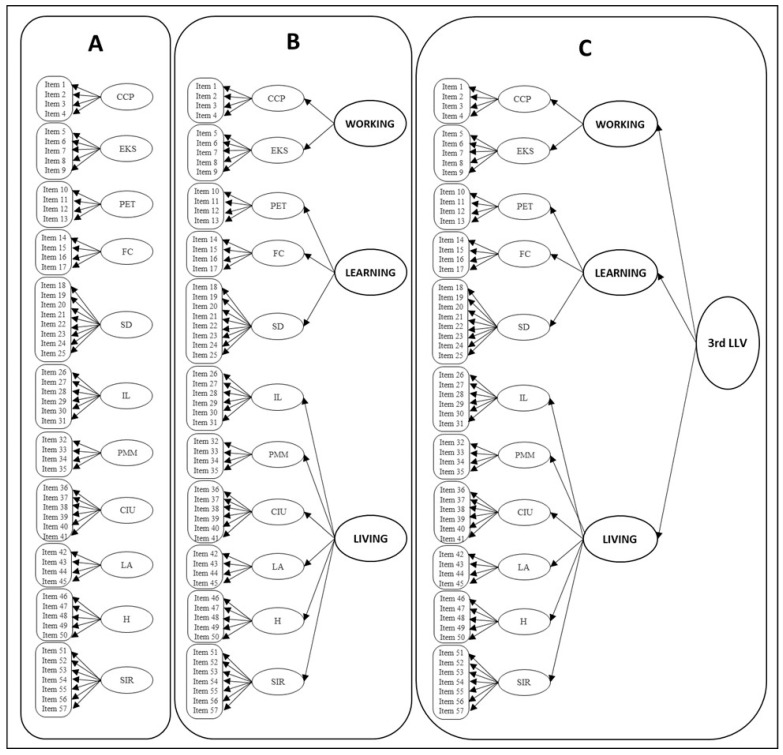
Three models of TPI-AR: (**A**) 1st-order model with eleven separated scales; (**B**) 2nd-order model with 3 higher latent variables; (**C**) 3rd-order model with higher overall latent variable. 3rd LLV = 3rd-level latent variable, CCP = Career Choice and Planning, EKS = Employment Knowledge and Skills, PET = Postsecondary Education/Training, FC = Functional Communication, SD = Self-Determination, IL = Independent Living, PMM = Personal Money Management, CIU = Community Involvement and Usage, LA = Leisure Activities, H = Health, SIR = Social/Interpersonal Relationships.

**Table 1 ijerph-20-01135-t001:** Participant Characteristics.

		N	%
Students			
Gender	Male	53	26
	Female	150	74
Disability	Intellectual disability	134	66
	Learning disabilities	22	11
	Hearing impairment	31	15
	Visual impairment	16	8
Age	13–15	77	38
	16–18	70	35
	19–21	31	15
	22+	25	12
School	General education	41	20
	Special education	125	62
	Day care center	37	18
Family (Parents)			
Gender	Male	58	29
	Female	145	71
Relative	Mother	99	49
	Father	44	22
	Sister	41	20
	Brother	14	7
	Other	5	2
Education	Uneducated	16	8
	High school	100	49
	Bachelor’s degree	80	40
	Postgraduate degree	7	3
Teachers ^a^			
Gender	Male	51	25
	Female	150	74
Experience	1–5	43	21
	6–10	99	49
	11+	59	29
Qualification	Diploma	5	3
	Bachelor’s degree	163	80
	Postgraduate degree	33	16

^a^ Two missing data.

**Table 2 ijerph-20-01135-t002:** Cronbach’s alpha for all scales.

Planning Areas Inventory	Pilot Sample (*n* = 30)			Full Sample (*n* = 203)			Number of Items
	Student	Home	School	Student	Home	School	
Working							9
Career Choice and Planning	0.528	0.889	0.794	0.884	0.899	0.928	4
Employment Knowledge and Skills	0.574	0.937	0.944	0.872	0.946	0.943	5
Learning							16
Further Education/Training	0.767	0.886	0.876	0.881	0.922	0.914	4
Functional Communication	0.745	0.899	0.903	0.845	0.918	0.922	4
Self-Determination	0.872	0.943	0.955	0.939	0.959	0.970	8
Living							32
Independent Living	0.542	0.833	0.896	0.887	0.920	0.940	6
Personal Money Management	0.658	0.855	0.801	0.838	0.888	0.894	4
Community Involvement and Usage	0.725	0.863	0.868	0.889	0.914	0.930	6
Leisure Activities	0.502	0.930	0.924	0.839	0.933	0.934	4
Health	0.709	0.899	0.884	0.873	0.929	0.942	5
Interpersonal Relationships	0.596	0.916	0.865	0.903	0.909	0.939	7

**Table 3 ijerph-20-01135-t003:** Fit indices for all scales for Model 1.

Planning Areas Inventory	CMIN	df	P	RMSEA	CFI	GFI	AGFI	TLI	NFI
Career Choice and Planning	*3.849*	*2*	*0.146*	*0.068*	*0.996*	*0.990*	*0.952*	*0.988*	*0.992*
	**0.062**	**1**	**0.804**	**0.000**	**1.000**	**1.000**	**0.998**	**1.010**	**1.000**
	5.770	1	0.016	0.154	0.994	0.986	0.860	0.961	0.992
Employment Knowledge and Skills	*6.062*	*4*	*0.195*	*0.051*	*0.996*	*0.988*	*0.954*	*0.991*	*0.990*
	**15.878**	**4**	**0.003**	**0.121**	**0.989**	**0.972**	**0.894**	**0.972**	**0.985**
	8.988	4	0.061	0.079	0.996	0.983	0.935	0.989	0.992
Further Education/Training	*10.545*	*2*	*0.005*	*0.145*	*0.981*	*0.975*	*0.873*	*0.942*	*0.976*
	**0.006**	**1**	**0.938**	**0.000**	**1.000**	**1.000**	**1.000**	**1.009**	**1.000**
	9.887	1	0.002	0.211	0.986	0.976	0.765	0.915	0.984
Functional Communication	*0.708*	*1*	*0.400*	*0.000*	*1.000*	*0.988*	*0.983*	*1.004*	*0.999*
	**0.231**	**1**	**0.631**	**0.000**	**1.000**	**0.999**	**0.994**	**1.006**	**1.000**
	21.379	1	0.000	0.319	0.975	0.952	0.518	0.848	0.974
Self-Determination	*82.792*	*19*	*0.000*	*0.129*	*0.954*	*0.909*	*0.828*	*0.932*	*0.942*
	**46.104**	**17**	**0.000**	**0.092**	**0.983**	**0.948**	**0.891**	**0.972**	**0.974**
	58.633	18	0.000	0.106	0.981	0.931	0.862	0.971	0.973
Independent Living	*18.017*	*8*	*0.021*	*0.079*	*0.985*	*0.972*	*0.927*	*0.971*	*0.973*
	**19.557**	**7**	**0.007**	**0.094**	**0.986**	**0.968**	**0.905**	**0.970**	**0.979**
	43.974	7	0.000	0.163	0.968	0.935	0.805	0.932	0.963
Personal Money Management	*19.532*	*2*	*0.000*	*0.208*	*0.946*	*0.960*	*0.800*	*0.838*	*0.941*
	**1.636**	**1**	**0.201**	**0.056**	**0.999**	**0.996**	**0.960**	**0.993**	**0.997**
	1.903	1	0.168	0.067	0.999	0.995	0.953	0.991	0.997
Community Involvement and Usage	*11.596*	*7*	*0.115*	*0.057*	*0.994*	*0.981*	*0.944*	*0.986*	*0.984*
	**18.660**	**6**	**0.005**	**0.102**	**0.986**	**0.972**	**0.901**	**0.965**	**0.980**
	17.084	7	0.017	0.085	0.991	0.974	0.923	0.981	0.985
Leisure Activities	*1.131*	*2*	*0.568*	*0.000*	*1.000*	*0.997*	*0.986*	*1.008*	*0.996*
	**6.796**	**2**	**0.033**	**0.109**	**0.993**	**0.983**	**0.917**	**0.979**	**0.990**
	8.551	1	0.003	0.194	0.990	0.980	0.795	0.939	0.989
Health	*11.307*	*4*	*0.023*	*0.095*	*0.988*	*0.978*	*0.917*	*0.970*	*0.981*
	**7.314**	**3**	**0.063**	**0.084**	**0.995**	**0.986**	**0.930**	**0.984**	**0.992**
	3.962	3	0.266	0.040	0.999	0.992	0.961	0.997	0.996
Interpersonal Relationships	*27.836*	*12*	*0.006*	*0.081*	*0.982*	*0.961*	*0.910*	*0.969*	*0.969*
	**27.226**	**9**	**0.001**	**0.100**	**0.983**	**0.966**	**0.894**	**0.961**	**0.976**
	32.032	10	0.000	0.105	0.985	0.957	0.879	0.969	0.979

Student form = *italicized script*, home form = bold script, school form = regular script, CMIN = Chi-square value, df = degree of freedom, CFI = comparative fit index; TLI = Tucker-Lewis index, GFI = goodness of fit, AGFI = adjusted goodness of fit, RMSEA, root mean square error of approximation; NFI = Normed Fit Index.

**Table 4 ijerph-20-01135-t004:** Composite reliability and average variance extracted for Model 1.

Sub-Scale	CR			AVE		
	Student	Parent	Teacher	Student	Parent	Teacher
Career Choice and Planning	0.85	0.87	0.92	0.67	0.71	0.78
Employment Knowledge and Skills	0.81	0.93	0.93	0.59	0.79	0.77
Further Education/Training	0.79	0.88	0.85	0.66	0.75	0.73
Functional Communication	0.77	0.89	0.88	0.59	0.74	0.73
Self-Determination	0.92	0.95	0.96	0.67	0.75	0.81
Independent Living	0.83	0.89	0.93	0.58	0.66	0.74
Personal Money Management	0.74	0.86	0.86	0.56	0.69	0.69
Community Involvement and Usage	0.83	0.88	0.90	0.58	0.64	0.69
Leisure Activities	0.75	0.89	0.90	0.57	0.78	0.78
Health	0.84	0.91	0.93	0.62	0.73	0.77
Interpersonal Relationships	0.86	0.86	0.93	0.59	0.59	0.71

CR = Composite Reliability, AVE = Average Variance Extracted.

## Data Availability

The data will be available upon request to the corresponding author g.alnahdi@psau.edu.sa.

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
