# Peer review of "Validation of the Arabic Version of the Transition Planning Inventory (TPI-AR)"

_ijerph, 2023, doi:10.3390/ijerph20021135_

Round 1

Reviewer 1 Report

This is  a very important and relevant paper in the field of Inclusive education. The study is rigorous and comprehensive and the paper is clear and well written. The authors clearly describe their methodology and reflect on their limitations. They also identify clearly implications for practice and future research. I would support publishing the manuscript with only a couple of minor suggestions

1. I would call the fist section of the manuscript Introduction or Background. 

2. The authors may want to clarify what '"transitional ages" means, the authors do explain it, when they explain the tool but it may be useful to explain it from the start,  in their sentence " Individuals in transitional stages have also 23 been a part of this focus in terms of supportive regulations."

3. The manuscript is well written but will benefit from checking for minor spelling errors. 

Author Response

Thank you for your feedback and your time. 

  1.  the first section of the manuscript was changed to "Introduction" as you suggested.  
  2.  We have added an additional sentence in the introduction to clarify the  '"transitional ages" meaning. 
  3.  We have checked and corrected some spelling errors as you suggested. 

Thank you 

Reviewer 2 Report

The manuscript is clear, relevant to the field of special education. The topic presented is timely and useful for providing a quality transition program for students with disabilities in Arab countries. The purpose of the paper is clearly stated, it is to see if the Arabic version of the TPI-AR is a valid and reliable tool for use in Saudi Arabia. The need to address the topic is justified by the lack in the Arab world of an appropriate tool for planning the transition to life after school for students with disabilities. This lack hinders the implementation of governmental directions to support transitions, as well as the concerted efforts teachers, professionals and parents make to monitor and support the process.  The manuscript is presented in a well-structured manner. In addition to a clear rationale for the need to address the topic, the introduction provides the necessary information on the construction and psychometric properties of the versions of the tool used to date, its use in other countries, and its compliance with IDEA. It is only advisable to include a brief description of the regulations governing the transition program in Saudi Arabia (23-24) to get a perspective of what the transition process looks like there. The theoretical overview is followed by a description of the method, results, their discussion, conclusions and bibliography. The manuscript thus contains all the required elements. Participant characteristics and the data collection process are described in detail. The development of the Arabic version of the tool followed appropriate procedures, including the approval of the TPI authors, the involvement of independent translators, a back-translation procedure, and assurances that the original meaning of the items would be preserved. The study was preceded by a pilot study. Appropriate statistical analyses (Cronbach's alpha, multivariate confirmatory analysis) were used to test the hypothesis, in addition, their use in studies by other authors was presented. The description of the studied models is presented in a clear manner. The figure and tables are easy to interpret and understand.  The data are interpreted in an appropriate and consistent manner. For greater readability, interpretations of selected detailed results are included.  A clear description of the procedure allows the results to be reproduced. The discussion points out the numerous benefits to practice and the broad applicability of the tool to different groups of students with disabilities educated in different types of schools. The analytical results presented in the manuscript indicate the applicability of each scale separately , which has important implications for both research and practice. In addition, the applicability of the Multitrait-Multi- Method (MTMM) in future research is explained. The conclusions are consistent with the evidence and arguments presented. Most of the cited references (34) are not recent publications (within the last 5 years), but relevant to the topic.  The number of self-citations is very limited.

Author Response

Thank your comments and positive feedback. We really appreciate your time and effort to read our paper. Thank you again